# *ANXA11* rs1049550 Associates with Löfgren’s Syndrome and Chronic Sarcoidosis Patients

**DOI:** 10.3390/cells11091557

**Published:** 2022-05-05

**Authors:** Bekir Karakaya, Joanne J. van der Vis, Marcel Veltkamp, Douwe H. Biesma, Jan C. Grutters, Coline H. M. van Moorsel

**Affiliations:** 1Interstitial Lung Diseases Center of Excellence, Department of Pulmonology, St. Antonius Hospital, Mailbox 2500, 3430 EM Nieuwegein, The Netherlands; a.vandervis@antoniusziekenhuis.nl (J.J.v.d.V.); m.veltkamp@antoniusziekenhuis.nl (M.V.); j.grutters@antoniusziekenhuis.nl (J.C.G.); c.van.moorsel@antoniusziekenhuis.nl (C.H.M.v.M.); 2Interstitial Lung Diseases Center of Excellence, Department of Clinical Chemistry, St. Antonius Hospital, Mailbox 2500, 3430 EM Nieuwegein, The Netherlands; 3Division of Heart & Lungs, University Medical Center Utrecht, Postbus 85500, 3508 GA Utrecht, The Netherlands; 4Department of Internal Medicine, Leiden University Medical Center, Mailbox 9600, 2300 RC Leiden, The Netherlands; d.h.biesma@lumc.nl

**Keywords:** sarcoidosis, Löfgren’s syndrome, ANXA11, SNP

## Abstract

Sarcoidosis is an immune mediated granulomatous disease commonly affecting the lungs. Genome wide association studies identified many genomic regions that are shared among multiple immune mediated diseases. However, *ANXA11* gene polymorphism rs1049550 is exclusively associated with sarcoidosis, making it a key gene of interest for sarcoidosis disease pathogenesis. However, sarcoidosis is a heterogeneous disease and contradictory findings for ANXA11 have been reported for disease phenotypes. We performed a case–control association study to investigate if *ANXA11* associates with benign (Löfgren’s syndrome (LS)) or chronic sarcoidosis and performed a meta-analysis on previously reported findings. A total of 262 sarcoidosis patients, of which 149 had LS and 113 chronic sarcoidosis, and 363 controls were genotyped for rs1049550. Meta-analysis included allele findings for rs1049550 from 6 additional studies. We found a significantly lower T allele frequency in sarcoidosis patients than in healthy controls (0.30 vs. 0.41, respectively, odds ratio (OR) 0.61, 95% confidence interval (CI) 0.48–0.77, *p* = 3 × 10^−5^). In LS the T allele frequency of 0.33, and in chronic sarcoidosis the T allele frequency of 0.26 were significantly lower than in healthy controls (OR 0.69, 95% CI 0.52–0.92, *p* = 0.01 and OR 0.51, 95% CI 0.36–0.70, *p* = 4 × 10^−5^, respectively). Meta-analysis including previously published European, African American and Asian cohorts confirmed the association of rs1049550 with sarcoidosis and resulted in a pooled OR of 0.70 (CI 0.66–0.75, *p* = 3.58 × 10^−29^). Presence of the T allele of rs1049550 in *ANXA11* is protective for sarcoidosis, including benign and chronic phenotypes of the disease.

## 1. Introduction

Sarcoidosis is a systemic inflammatory disorder of unknown cause with a wide clinical spectrum. The disease is characterized by the formation of non-caseating epithelioid cell granulomas. It commonly affects the lungs and intrathoracic lymph nodes, but any organ can be involved. The clinical course and prognosis is heterogeneous. Löfgren’s syndrome (LS) is a well-defined phenotype of sarcoidosis manifesting as acute onset, bilateral hilair lymphadenopathy (BHL), erythema nodosum, and/or bilateral ankle arthritis or periarticular inflammation [1]. Patients with LS usually have a very good prognosis in contrast with the non-LS patients of whom about one third develop chronic sarcoidosis [2].

Sarcoidosis is a complex disease, suggested to be the result of an interaction between an environmental trigger and a patient’s genetic make-up. Genome wide association studies (GWAS) showed that multiple genes associate with sarcoidosis, and most of these gene regions are shared among multiple immune mediated diseases. However, the association between *ANXA11* and sarcoidosis, which was for the first time described in a German cohort by Hofmann et al. [3] is unique for sarcoidosis. Several studies have replicated these results and showed an association with the *ANXA11* Single Nucleotide Polymorphism (SNP) rs1049550 and sarcoidosis [4,5,6,7,8,9]. All studies found that carriage of the minor T allele conferred protection for sarcoidosis, as the minor T allele frequency was lower in sarcoidosis patients than in controls. However, subsequent subgroup analyses investigated whether this SNP associated with specific disease phenotypes of sarcoidosis, resulting in some contradictory findings when groups were formed according to radiological Scadding stages. Where one study found a lower T allele frequency in Scadding stage II–IV patients [5], two other studies [8,10] found a higher T allele frequency in Scadding stage II–IV patients when compared to a group of patients with lower Scadding stages. 

Furthermore, two studies described findings in a sarcoidosis subgroup consisting of LS patients. Mrazek et al. [5] found a higher TT frequency in LS patients compared to non-LS patients, and Morais et al. [7] found no difference between LS patients and controls regarding rs1049550. However, both studies had limited power and thus wide confidence intervals, as the numbers of LS patients in both studies were relatively small, with 39 and 55 patients, respectively. 

Most genes identified by GWAS do not only associate with sarcoidosis, but have also been identified in studies for immune-mediated disorders, such as inflammatory bowel disease and Crohn’s disease [11]. *ANXA11* is currently the only gene identified by GWAS that predisposes to sarcoidosis but not to other immune mediated diseases [12]. *ANXA11* may therefore provide the key to understanding sarcoidosis-specific disease processes. However, while the association between sarcoidosis in general and rs1049550 has been confirmed in several studies, the protective role of the rs1049550 T allele for clinical phenotypes of sarcoidosis remains to be determined. Prior to further experimental studies into the role of ANXA11 in sarcoidosis, it is essential to determine for which phenotypes the T allele may protect. The aim of this study was therefore, first, to investigate the association of the *ANXA11* SNP rs1049550 with sarcoidosis in general and corroborate previous findings by adding new results in a meta-analysis. Second, to analyze clinical sub-phenotypes LS and chronic sarcoidosis to provide new evidence about the effect of rs1049550 T allele. These data will provide insights on the role of *ANXA11* polymorphism in sarcoidosis in general and specific disease phenotypes and may influence experimental design and choice of therapy in the future.

## 2. Materials and Methods

### 2.1. Subjects

A total of 262 Dutch sarcoidosis patients from St. Antonius Hospital, Nieuwegein, the Netherlands, were included in the study. All patients were diagnosed in accordance with the consensus of the ATS/ERS/WASOG statement on sarcoidosis [13].

LS was defined as presenting with the classic symptoms: acute onset with bilateral hilar lymphadenopathy, fever, erythema nodosum (EN), and/or bilateral ankle arthritis [1]. 

Thoracic involvement was classified using the Scadding criteria [14]: stage 0, no lung involvement; stage I, lymphadenopathy without parenchymal involvement; stage II, lymphadenopathy with parenchymal involvement; stage III, parenchymal involvement only; and stage IV, pulmonary fibrosis. 

Our group consisted of 149 patients with LS and 113 patients with chronic sarcoidosis. Patients with chronic disease had evidence of disease after at least 4 years of follow-up. 

Three hundred and sixty-three healthy Caucasian subjects were included as controls in this study.

Written consent was obtained from all subjects, and authorization was given by the Medical research Ethics Committees United (MEC-U) of the St. Antonius Hospital, Nieuwegein (approval number R08-37A).

### 2.2. Genotyping

Genomic DNA was extracted from peripheral blood of each individual using standard methods. *ANXA11* rs1049550 was genotyped with a pre-designed taqman SNP genotyping assay (Assay ID C_7881261_1) and the Quantstudio^®^ 5 real-time PCR system (both ThermoFisher Scientific, Waltham, MA, USA).

### 2.3. Meta-Analysis

For meta-analysis, we included studies with (a) a case–control study design, (b) diagnosis of sarcoidosis according to internationally accepted criteria, and (c) reporting genotype or allele frequencies. For the study by Levin et al. [6], we switched the C and T alleles (see discussion). For the study by Feng et al. [8], we used published frequencies for the first meta-analysis and excluded the study in a second meta-analysis.

Most of the studies included in the first meta-analysis also analyzed phenotypes of sarcoidosis. We also performed a meta-analysis for the phenotypes of sarcoidosis, resolving and chronic sarcoidosis, LS and non-LS, by using the data provided in the studies. In our LS patients, there were 7 patients with chronic disease. By excluding these 7 patients from our LS patients, we had a group of patients with resolving disease, which we included for the meta-analysis. We did not perform a meta-analysis with the Scadding stage data due to the variation in selected Scadding stages. However, the data about Scadding stages and *ANXA11* rs1049500 are presented in Appendix A.

### 2.4. Statistical Analysis

Allele and genotype frequencies were calculated for SNP rs1049550 and tested for Hardy–Weinberg equilibrium (HWE). Differences in allele frequencies were calculated with the Pearson’s goodness-of-fit Chi-square test. Three genetic models (additive, dominant, and recessive) were assessed with a logistic regression analysis on genotype results using SNPStats [15]. The odds ratio (OR) and confidence interval (95% CI) were calculated, and a *p* value < 0.05 was considered statistically significant. 

Furthermore, when presenting data from previous studies, the OR, 95% CI, and *p*-value were calculated from the data described in the original publication; when these data were not available the OR, 95% CI, and *p*-values were copied from the original article. 

Meta-analyses were performed using the allele contrast in the web tool META-Genyo [16]. Heterogeneity in the data was evaluated with I^2^ statistics and Cochran’s Q test was low for the allele contrast. The fixed-effect estimate method inverse variance was used. Publication bias was investigated by Egger’s regression test. 

## 3. Results

### 3.1. Sarcoidosis Patients and Subgroups

In total, 262 sarcoidosis patients and 363 healthy controls were included. The patient cohorts consisted of 149 patients with LS and 113 non-LS patients with chronic sarcoidosis and Scadding stage II–IV (Table 1). Patients with LS were more often female and chronic sarcoidosis patients were more often male and slightly older.

### 3.2. Association of ANXA11 rs1049550 with Sarcoidosis

Genotyping results are shown in Table 2. For all groups, no deviation from Hardy–Weinberg equilibrium was observed (*p* > 0.05). A significantly decreased minor T allele frequency was observed in the total group of sarcoidosis patients compared with controls (0.30 vs. 0.41, OR 0.61, 95% CI 0.48–0.77, *p* = 3 × 10^−5^). 

Subgroup analysis showed that comparison of LS with controls yielded a significantly lower minor T allele frequency of 0.33 in LS with an OR 0.69 (95% CI 0.52–0.92, *p* = 0.01). Additionally, comparison of chronic sarcoidosis patients with controls also yielded a significantly lower minor T allele frequency of 0.26 with an OR 0.51 (95% CI 0.36–0.70, *p* = 4 × 10^−5^; Table 2). Furthermore, no significant difference was found between the LS and chronic sarcoidosis groups.

In addition to the allelic analysis, the association between *ANXA11* rs1049550 and total group of sarcoidosis patients, LS, and chronic sarcoidosis patients was assessed for underlying genetic model (Table 3). The additive and recessive (CC+CT vs. TT) models provide comparable significant results in sarcoidosis and its phenotypes. No significant results were retrieved for the dominant (CC vs. CT+TT) model in LS.

### 3.3. Meta-Analysis

For meta-analysis, we included six previous studies [3,4,5,6,7,8] describing the association between ANXA11 rs1049550 and sarcoidosis and the current findings (Table 4 and Figure 1). One study [9] that did not describe the genotype frequencies was added to Table 4 to provide a complete overview of studies performed, but could not be included in the meta-analysis. The meta-analysis showed that sarcoidosis patients have a significantly lower T allele frequency in comparison to controls, with a pooled OR of 0.70 (95% CI 0.66–0.75, *p* = 3.58 × 10^−29^). There was no significant heterogeneity across the studies, and no significant publication bias was detected by Egger’s test, *p* = 0.52. As the T allele frequency of the control cohort in Feng and co-authors [8] deviated significantly from that in public databases, we also performed a meta-analysis without that study, obtaining a pooled OR of 0.71 (95% CI 0.67–0.76, *p* = 2.03 × 10^−24^).

### 3.4. Sarcoidosis Phenotype Studies

In Table 5, we show the comparisons made in previous studies [4,5,6,7] and the present study between phenotypes of sarcoidosis: four studies for chronic and resolving disease and three studies for LS. A summary of all studies [4,5,6,7,8,9,10] is provided in the Appendix A for comparison of reported ORs and in Appendix A for Scadding stage data. All phenotype studies found lower T allele frequency for phenotypes when compared with controls, however in some studies this was not significant. 

Combining previous studies and new data, we performed meta-analyses for the following phenotypes of sarcoidosis: resolving disease and chronic disease, LS and non-LS. Levin et al. [6] and Mrazek et al. [5] did not provide genotype data, so these studies were not incorporated in the meta-analysis for the phenotypes of sarcoidosis.

Pooled data show a significantly decreased T allele frequency in resolving disease, chronic disease, LS, and non-LS when compared with controls, with an OR of 0.65 (95% CI 0.54–0.78, *p* = 4.7 × 10^−6^), an OR of 0.62 (95% CI 0.51–74, *p* = 3.8 × 10^−7^), an OR of 0.69 (95% CI 0.55–0.88, *p* = 0.0025), and an OR of 0.54 (95% CI 0.43–0.68, *p* = 1.1 × 10^−7^), respectively (Figure 2A,B). Comparison between resolving and chronic disease, or LS and non-LS phenotypes, did not yield significant results.

Comparison between Scadding stages and controls showed similar results (Appendix A), however comparing Scadding stages with each other showed contradictory findings. It is important to note that the compared Scadding stages differed from each other in the studies, i.e., where one study included patients with Scadding stage II–IV in the higher stage group, the other included only patients with Scadding stage IV.

## 4. Discussion

In the present study, we investigated if *ANXA11* rs1049550 associates with sarcoidosis and with disease phenotypes. Meta-analysis shows that the SNP rs1049550 associates with sarcoidosis resulting in a pooled OR of 0.70 (95% CI 0.66–0.75); *p* = 3.58 × 10^−29^. Furthermore, we show that this association is independent of sarcoidosis phenotype, because in both LS and chronic sarcoidosis a similar association was present.

The association between rs1049550 and sarcoidosis was described in a German cohort for the first time by Hofmann et al. [3], who performed the first genome-wide association study (GWAS) in sarcoidosis. Afterwards, rs1049550 was studied in other populations of sarcoidosis and phenotypes of the disease [4,5,6,7,8,9,10]. However, due to corrigenda and contradictory findings, especially regarding the clinical phenotypes that were studied, the results have been difficult to oversee. The GWAS report was followed by a correction of the notation of the SNP, reversing the minor allele from c.688C into c.688T [17]. Following the GWAS study, several studies investigated this association and correctly used the T allele as the minor allele. However, in a study by Levin et al. [6], the C allele was again described as the minor allele in an European American and an African American population. This description not only contradicted previous findings, but also with reported allele frequencies in public databases [18]. Most confusing, the authors concluded that their results were in line with the study by Hofmann et al. [3], which would mean that the alleles were switched. We have therefore chosen to switch these alleles in our meta-analysis and forest plots, resulting in the T allele being the minor allele. In 2014, a meta-analysis [19] was performed on six studies, including the study of Levin et al. [6], but without correction for the proper minor allele and thus with an inversely calculated OR. 

Furthermore, to determine if control cohorts conferred to established allele frequencies in ethnic cohorts, we checked rs1049550 in the genome aggregation database (gnomAD) [18]. We observed that the T allele frequency varies considerably between populations, with the lowest frequency in African/African Americans (0.1994), with a slightly higher frequency in South Asian (0.3592) and in a European population (0.4206). However, in East Asian populations of gnomAD (v3.1.2) [18], the frequency of the T allele is reported to be considerably higher at 0.6633. Within gnomAD, the 1000 genome (1 KG) project shows specific East Asian findings: for Southern Han Chinese and Han Chinese in Beijing, there are reported T allele frequencies of 0.6619 and 0.6373, respectively. Surprisingly, Feng et al. [8] report a Chinese control population from Xinxiang (a city about 600 km to the southwest of Beijing) with a T allele frequency of 0.40. As this is far from the database report of East Asian frequencies, further research is needed to determine the correct control frequency and compare this with a population of sarcoidosis patients from the same region. Although genetic studies in different ethnic populations can be difficult to compare, the case–control meta-analysis for rs1049550 yielded a strong significant result. For our meta-analysis, we incorporated the reported findings of Feng et al. [8]; if we exclude these findings a highly similar result was found. 

In several studies, associations between rs1049550 and phenotypes of sarcoidosis were investigated (Table 5 and Figure 2). These phenotypes are partly, but not completely, overlapping. LS is usually non-chronic/resolving with Scadding stage 0 or I, whereas chronic pulmonary disease predominantly involves Scadding stages II and up. Most studies comparing disease phenotypes with controls found that the T allele protected against the phenotype of sarcoidosis. The T allele frequencies in resolving and in chronic disease were significantly lower than in controls [4,6] (Table 5 and Figure 2A). Furthermore, no difference was found between resolving (acute) and chronic sarcoidosis [4,7].

LS is one of the best studied phenotypes of sarcoidosis. This phenotype is characterized by specific genetic associations, like with HLA-DR3 [20], CCR5 [21], and CCR2 [22], that are not all shared with non-LS sarcoidosis. Data for *ANXA11* in LS are scarce; only two studies mention patients with LS; Mrazek et al. [5] included 39 LS patients. Data about the T allele frequency in LS patients were not provided, but they reported a significantly higher frequency of the TT genotype when compared to non-LS patients. However, TT genotype frequency in LS patients was still lower than in controls, but no comparison was made. Morais et al. [7] did not find any difference between LS (*n* = 55) and non-LS patients in T allele frequency. We therefore analyzed rs1049550 in a large cohort of 149 Dutch LS patients, all with Scadding stage 0 or I except for nine patients with Scadding stage II. Our results demonstrated a similar association of rs1049550 with LS, as seen in sarcoidosis patients, which is a protective effect of the minor T allele. Furthermore, we showed that there is no difference between LS and chronic sarcoidosis. A meta-analysis with the pooled data from Morais et al. [7] and our data shows a lower T allele frequency in LS patients compared to controls, but no difference when compared to non-LS patients (Figure 2B). Therefore, we can conclude that the association of *ANXA11* rs1049550 applies for sarcoidosis in general and the clinical phenotypes of LS and chronic disease. Presumably, the association may be found in all sarcoidosis cohorts, regardless of phenotype.

In the past years, several studies found genetic commonalities between sarcoidosis and other immune modulated diseases, such as Crohn’s disease [23]. However, for *ANXA11*, no associations with other immune modulated diseases are published. The *ANXA11* rs1049550 polymorphism leads to an amino-acid substitution of arginine to cysteine, at position 230 (p.(R230C)) of the Annexin A11 protein, within a highly conserved domain responsible for Ca^2+^-binding properties [24]. Annexin A11 is known to be involved in calcium signaling, cell cycle, vesicle trafficking, and apoptosis [25,26]. The effect of this amino-acid change is not well defined; although Hofmann et al. [3] showed no difference in ANXA11 mRNA expression between control and diseased lung tissue, they showed a downregulation of *ANXA11* mRNA expression in CD8+T and CD19+B cells after stimulation. Furthermore, ANXA11 mRNA expression in bronchoalveolar lavage (BAL) cells and peripheral blood mononuclear cells (PBMC) was not associated with rs1049550 genotype [5]. However, the Open Targets Genetics [27] database shows imputed results for rs1049550, and finds a strong association with sarcoidosis and differential expression of ANXA11 for the lead variant. This strongly suggest that ANXA11 is indeed the quantitative trait for rs1049550. Moreover, this differential expression is observed in monocytes [28]. 

Levin et al. [6] found a significant SNP–SNP interaction between rs1049550 and a *HLA* SNP. This may indicate an interplay between HLA molecules, by presenting antigens, and ANXA11 by influencing the inflammatory response. Both ANXA11 and HLA molecules are among the few proteins detectable in human B cell exosomes. In BAL fluid of sarcoidosis patients, an increased level of exosomes was shown, especially expressing major histocompatibility complex class I and II proteins [29]. It is hypothesized that exosomes play a role in B cell activation thereby lowering the threshold for T cell activation [29]. It has been suggested that the amino-acid change caused by rs1049550 results in a dysfunctional Annexin A11 which can influence cell processes, thereby altering cell trafficking and apoptosis, which in turn can influence granuloma formation and maintenance, respectively, in sarcoidosis patients [30]. Further hypothesizing that without apoptosis the inflammatory and granulomatous inflammation will continue. There is one important aspect to take into account; the minor T allele, which would affect the Annexin A11 function, is actually less frequent in sarcoidosis and therefore its presence protects against sarcoidosis. 

Now that we established that *ANXA11* rs1049550 associates with sarcoidosis in general, regardless of disease phenotype, more research is needed to elucidate the effects of the gene variant on Annexin A11 function and how this effects granuloma formation and maintenance in sarcoidosis patients. On the subject level, we found that the genetic additive or recessive (CC+CT vs. TT) model may underlie the association with sarcoidosis and its phenotypes, suggesting that future experimental studies should study differences between TT and CC genotypes. It would be of great interest to investigate ANXA11 in cell types known to be involved in granuloma formation, such as macrophages and monocytes. Other interesting cells would be CD4+, CD8+, and CD19+ T cells, which are also known to play an important role in sarcoidosis.

## Figures and Tables

**Figure 1 cells-11-01557-f001:**
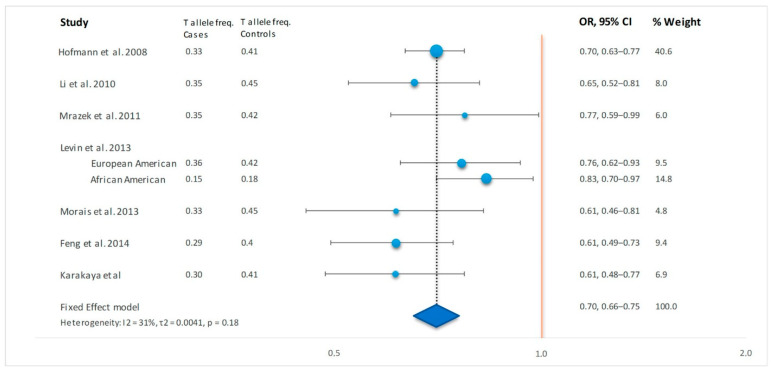
Meta-analysis for the studies of ANXA11 with sarcoidosis. Forest plot of the odds ratio for *ANXA11* rs1049550 T allele in sarcoidosis cases versus controls. For correct presentation, the C and T allele are switched for the studies of Levin et al. Dotted line represents the odds ratio from the combined analysis. The result of the meta-analysis is presented as a diamond at the bottom that covers the combined OR in the middle and the CI at the tips. OR: Odds ratio, 95% CI: 95% confidence interval.

**Figure 2 cells-11-01557-f002:**
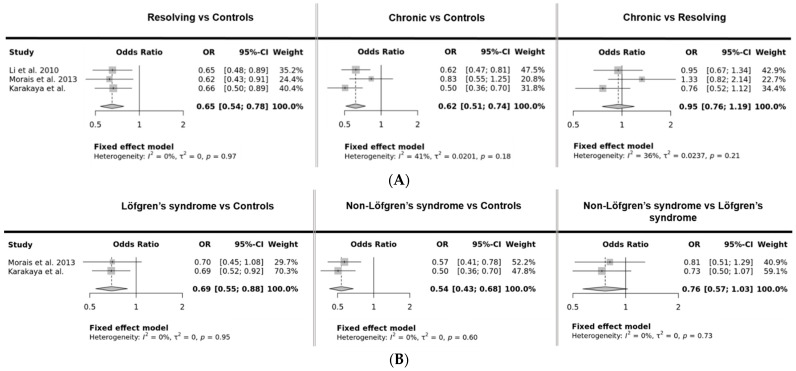
Meta-analysis for the studies of *ANXA11* rs1049550 and phenotypes of Sarcoidosis. (**A**) Forest plot of the results for resolving and chronic sarcoidosis patients. (**B**) Forest plot of the results for Löfgren’s syndrome and non-Löfgren’s syndrome patients. Dotted line represents the odds ratio from the combined analysis. OR: Odds ratio, 95% CI: 95% confidence interval.

**Table 1 cells-11-01557-t001:** Baseline characteristics of patients with Löfgren’s syndrome and chronic sarcoidosis.

		Controls	Sarcoidosis All	Löfgren’s Syndrome	Chronic Sarcoidosis
**N**		363	262	149	113
**Age** *yrs*		40	38	36	42
**Female** *n (%)*		185 (50.1)	130 (49.6)	94 (63.1)	36 (31.8)
**Scadding Stage**			*n* = 228	*n* = 115	*n* = 113
*n (%)*	0		5 (2.2)	5 (4.3)	
	I		101 (44.3)	101 (87.8)	
	II		38 (16.7)	9 (7.8)	29 (25.7)
	III		13 (5.7)		13 (11.5)
	IV		71 (31.1)		71 (62.8)

**Table 2 cells-11-01557-t002:** *ANXA11* rs1049550 genotype and allele frequencies of patients with Löfgren’s syndrome and chronic sarcoidosis.

		Controls	Sarcoidosis All	Löfgren’s Syndrome	Chronic Sarcoidosis
**N**		363	262	149	113
**Genotype**					
*n (%)*	CC	130 (35.8)	125 (47.7)	66 (44.3)	59 (52.2)
	CT	167 (46.0)	118 (45.0)	69 (46.3)	49 (43.4)
	TT	66 (18.2)	19 (7.3)	14 (9.4)	5 (4.4)
**Allele**					
*n (%)*	C	427 (58.8)	368 (70.2)	201 (67.4)	167 (73.9)
	T	299 (41.2)	156 (29.8)	97 (32.6)	59 (26.1)
**OR ***			0.61	0.69	0.51
**95% CI**			0.48–0.77	0.52–0.92	0.36–0.70
** *p* **			3 × 10^−5^	0.01	4 × 10^−5^

* comparison of the patients group with controls, OR, CI and *p*-value are related to the allelic model.

**Table 3 cells-11-01557-t003:** Analysis of genetic models possibly underlying the association between *ANXA11* rs1049550 and sarcoidosis.

		Additive		Dominant (CC vs. CT+TT)		Recessive (CC+CT vs. TT)	
	*n*	OR, 95% CI *	*p* Value	OR, 95% CI *	*p* Value	OR, 95% CI *	*p* Value
**Controls**	363						
**Sarcoidosis**	262	0.61, 0.48–0.77	<0.0001	0.61, 0.44–0.85	0.0029	0.35, 0.21–0.60	<0.0001
**Löfgren’s syndrome**	149	0.69, 0.52–0.92	0.01	0.70, 0.48–1.03	0.074	0.47, 0.25–0.86	0.0095
**Chronic Sarcoidosis**	113	0.51, 0.36–0.71	<0.0001	0.51, 0.33–0.78	0.002	0.21, 0.08–0.53	0.0001

* comparison of the patients group with controls.

**Table 4 cells-11-01557-t004:** Previous and present *ANXA11* rs1049550 association studies with sarcoidosis.

Author	Year	Country	Ethnicity	Case/Control, N	Case T Allele Freq	Control T Allele Freq	OR, 95% CI, *p* Value *	Population T Allele Frequency gnomAD
**Hofmann S et al.** [3]	2008	Germany	Caucasian	1636/1811	0.33	0.41	0.70, 0.63–0.77, 1 × 10^−12^	European: 0.42
**Li et al.** [4]	2010	Germany	Caucasian	349/313	0.35	0.45	0.65, 0.52–0.81, 1 × 10^−4^	European: 0.42
**Mrazek et al.** [5]	2011	Czech	Caucasian	245/254	0.35	0.42	0.77, 0.59–0.99, 0.04	European: 0.42
**Levin et al.** [6] **^#^**	2013	USA	European American	446/350	0.36	0.42	0.76, 0.62–0.93, 0.008	European: 0.42
			African American	1232/893	0.15	0.18	0.83, 0.70–0.97, 0.02	African/African American: 0.20
**Morais et al.** [7]	2013	Portugal	Caucasian	208/197	0.33	0.45	0.61, 0.46–0.81, 6 × 10^−4^	European: 0.42
**Feng et al.** [8]	2014	China	Chinese-Han	412/418	0.29	0.40	0.60, 0.49–0.73, 8 × 10^−7^	East Asian: 0.66 ^§^
**Sikorova et al.** [9]	2020	Greece	Caucasian	103/100	Not available	Not available	0.59, 0.39–0.89, 0.01 ^†^	European: 0.42
**Karakaya et al.**		Netherlands	Caucasian	262/363	0.30	0.41	0.61, 0.48–0.77, 3 × 10^−5^	European: 0.42

* Odds ratio (OR), 95% confidence interval (CI), and *p*-values are calculated from the data provided in the original articles. ^#^ For correct presentation, the C and T allele are switched. ^§^ The gnomAD database shows that the T allele in East Asian is the major allele. Within the gnomAD database, the 1 KG shows different East Asian populations: Han Chinese in Beijing have a T allele frequency of 0.6373; Southern Han Chinese have a T allele frequency of 0.6619. ^†^ The genotype data were not available for the study by Sikorova et al.; values were copied from the original article.

**Table 5 cells-11-01557-t005:** Studies reporting an association of *ANXA11* rs1049550 with sarcoidosis disease phenotypes.

Study	T Allele Frequency			OR, 95% CI, *p* Value *		
	**Controls**	**Resolving**	**Chronic**	**Resolving vs. Controls**	**Chronic vs. Controls**	**Chronic vs. Resolving**
**Li et al, 2010** [4]	0.45 (*n* = 313)	0.35 (*n* = 117)	0.34 (*n* = 176)	0.65, 0.48–0.89, 0.007	0.62, 0.47–0.81, 5 × 10^−4^	0.95, 0.67–1.34, 0.76
**Levin et al. 2013** [6] **^#^**	0.18 (*n* = 893)	*n* = 304	*n* = 650	0.82, 0.64–1.06, 0.13 ^§,†^	0.79, 0.65–0.95, 0.02 ^§,†^	
**Morais et al. 2013** [7]	0.45 (*n* = 197)	0.34 (*n* = 86)	0.40 (*n* = 62)	0.62, 0.43–0.91, 0.01	0.83, 0.55–1.25, 0.37	1.33, 0.82–2.14, 0.24
**Karakaya et al.**	0.41 (*n* = 363)	0.32 (*n* = 142)	0.26 (*n* = 113)	0.66, 0.50–0.89, 0.005	0.51, 0.36–0.70, 4 × 10^−5^	0.76, 0.52–1.12, 0.65
	**Controls**	**Löfgren’s Syndrome**	**Non-Löfgren’s Syndrome**	**Löfgren’s Syndrome vs. Controls**	**Non-Löfgren’s Syndrome vs. Controls**	**Non-Löfgren’s Syndrome vs. Löfgren’s Syndrome**
**Morais, 2013** [7]	0.45 (*n* = 197)	0.36 (*n* = 55)	0.32 (*n* = 145)	0.70, 0.45–1.08, 0.11	0.57, 0.42–0.78, 5 × 10^−^^4^	0.81, 0.51–1.29, 0.38
**Karakaya et al.**	0.41 (*n* = 363)	0.33 (*n* = 149)	0.26 (*n* = 113)	0.69, 0.52–0.92, 0.01	0.51, 0.36–0.70, 4 × 10^−^^5^	0.73, 0.50–1.07, 0.11
**Mrazek, 2011** [5]	TT frequency: 0.15 (*n* = 254)	TT frequency: 0.21 (*n* = 39)	TT frequency: 0.07 (*n* = 147)			0.31, 0.11–0.84, 0.02 ^†^

* Odds ratio (OR), 95% confidence interval (CI), and *p*-values are calculated from the data provided in the original articles. ^#^ Study population: African Americans. For correct presentation, the C and T allele are switched. ^§^ The original article states that the additive genetic model was used to estimate the OR, and the OR is adjusted for sex and percent African ancestry. ^†^ Original data were not available; values are copied from the original article.

## Data Availability

The data presented in this study are available on request from the corresponding author.

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
