# Peer review of "ANXA11 rs1049550 Associates with Löfgren’s Syndrome and Chronic Sarcoidosis"

_cells, 2022, doi:10.3390/cells11091557_

Round 1
Reviewer 1 Report
In their manuscript, Karakaya and coauthors studied the association of rs1049550 with sarcoidosis, collecting and re-organizing results from literature and corroborating the variant effect on the disease with a meta-analysis including new data. They also analyzed the variant effect on several sarcoidosis sub-characterizations.
As a general comment, meta-analysis results, representing a replication/correction of the already reported association of rs1049550-T with protection against sarcoidosis, do not represent a major novelty: the same result has been confirmed also in large cohorts (although of European ancestry only), as reported in the Open Targets Genetics database:
https://genetics.opentargets.org/variant/10_80166946_G_A.
To me, the major novelty here is represented by the stratification of phenotypes, being a potentially interesting issue for clinical applications and, thus, should be better enhanced, as I would try to suggest below.
Several major comments (Ln=line):
- Ln 69: this sentence seems to be contradictory with that reported in Ln 51, where is declared that rs1049550 is associated with sarcoidosis and the association has been replicated in multiple studies. Moreover, the effect of the T allele is always in the same direction, conferring protection in all studies. Please better clarify here the aim of the paper.
- Ln 73: in my mind, the structure of this sentence (as well as the manuscript) should be:
- to evaluate the association of rs1049550 in new data and corroborate known results by adding new results in a meta-analysis.
- to analyze different clinical sub-phenotypes in order to provide new evidence about the effects (estimates and directions) of rs1049550-T and, consequently, inform on therapy.
This sentence should be modified. Some words related to the importance of it should be reported in the discussion too.
- Ln 138: OR, CI and P-value are related to the allelic model; it would be clearer if this is specified in the legend to avoid confusion with a genotypic model.
Did the authors test genotypic models (dominant, recessive, etc)? This information could be interesting and would be reported, even if the result is not significant.
- Ln 150 and 154 (Table3 and legend): authors reported rs1049550-T frequency is South Asians as 0.6619 while I found a frequency of 0.359 (from GnomAD v2.1.1:
https://gnomad.broadinstitute.org/variant/10-81926702-G-A?dataset=gnomad_r2_1),
please recheck this value.
The low allelic frequency observed in controls in Feng et al, more similar to South Asians than East Asians, should be due to any ascertainment bias, so excluding them from meta-analysis is a good point. Related to this, I think reporting somewhere heterogeneity statistics for the pooled OR =0.71 (Ln 149) would be important.
Also, I was wondering why Sikorova et al. 2020 study has not been included in this analysis, considering that the abstract reports:
“Further, ANXA11 rs1049550*A variant was associated with sarcoidosis (OR: 0.59 [0.39–0.89], p = 0.01).”
- Ln 174 Table 4: for several studies, no frequencies but only sample sizes are reported (e.g. in Sikorova et al. 2020): were these data not available in the original study?
In addition, it is not clear to me why in the Karakaya et al. results row, no comparison has been done between non-Löfgren’s syndrome patients and controls.
- Table 4 and Figure 2: these items include too many results and I am not sure a unique table/figure would be the best way to represent such different phenotypes.
Apart from this remark, I think when multiple results are present for the same phenotypes, a meta-analysis should be applied to obtain an overall measure of rs1049550-T clinical impact. Being this aspect the major novelty of this paper (rather than the overall association with sarcoidosis), I think these data (especially new data) should be capitalized to the maximum (unless the authors have valid scientific reasons against this observation) and, mainly, observations at Ln 265-266 in the discussion should be proven.
- Ln 273: molecular data from Open Targets Genetics show that ANXA11 is the first candidate gene regulated by rs1049550, with for example rs1049550-T that increases ANXA11 expression on monocytes and in other tissues:
https://genetics.opentargets.org/variant/10_80166946_G_A
The discussion could include some references to this aspect.
- Ln 288-291: this content seems not pertinent in the conclusions, although I think, as I pointed out before, the authors should better discuss possible implications of their results in a clinical context.
Minor comments:
- Ln 52: Table3 is the first table cited, thus should be named as “Table1”; please renumber tables.
- Ln 146 and 149, p < 1·10-10: I think the p-value should be more informative if its exact value is reported (either because corresponds to a narrower 95% CI and thus a more precise OR estimation with respect to the best p-value obtained in single studies [Hoffman, OR=0.70 and p-value = 1.10-12], and because coherent with the modest Heterogeneity identified).
- Ln 158 (Figure 1): as for the single study hits, the 95% CI should be plotted together with the combined OR.
Author Response
Response to reviewer #1
(x) I would not like to sign my review report
( ) I would like to sign my review report
English language and style
( ) Extensive editing of English language and style required
( ) Moderate English changes required
(x) English language and style are fine/minor spell check required
( ) I don't feel qualified to judge about the English language and style
|
Yes |
Can be improved |
Must be improved |
Not applicable |
|
|
Does the introduction provide sufficient background and include all relevant references? |
(x) |
( ) |
( ) |
( ) |
|
Is the research design appropriate? |
( ) |
(x) |
( ) |
( ) |
|
Are the methods adequately described? |
(x) |
( ) |
( ) |
( ) |
|
Are the results clearly presented? |
( ) |
( ) |
(x) |
( ) |
|
Are the conclusions supported by the results? |
( ) |
( ) |
(x) |
( ) |
Comments and Suggestions for Authors
In their manuscript, Karakaya and coauthors studied the association of rs1049550 with sarcoidosis, collecting and re-organizing results from literature and corroborating the variant effect on the disease with a meta-analysis including new data. They also analyzed the variant effect on several sarcoidosis sub-characterizations.
As a general comment, meta-analysis results, representing a replication/correction of the already reported association of rs1049550-T with protection against sarcoidosis, do not represent a major novelty: the same result has been confirmed also in large cohorts (although of European ancestry only), as reported in the Open Targets Genetics database:
https://genetics.opentargets.org/variant/10_80166946_G_A.
To me, the major novelty here is represented by the stratification of phenotypes, being a potentially interesting issue for clinical applications and, thus, should be better enhanced, as I would try to suggest below.
Several major comments (Ln=line):
- Ln 69: this sentence seems to be contradictory with that reported in Ln 51, where is declared that rs1049550 is associated with sarcoidosis and the association has been replicated in multiple studies. Moreover, the effect of the T allele is always in the same direction, conferring protection in all studies. Please better clarify here the aim of the paper.
Response: Indeed, the association of rs1049550 with sarcoidosis has been replicated in multiple studies, where the effect of the T allele was always found to be in the same direction. In most of these studies, sarcoidosis phenotypes were also investigated, and these yielded some contradictory findings suggesting that the rs1049550-T allele may not protect against all forms off sarcoidosis. For future (experimental) studies on this risk allele it is important to know in which phenotypes a protective effect may be observed. Our aim was therefore not only to confirm (and clearly tabulate, because there has been some miscommunication on the results) previous findings for sarcoidosis in general, but also to analyze clearly defined sarcoidosis phenotypes in order to provide new evidence for the role of ANXA11 in Löfgren’s syndrome and chronic sarcoidosis specifically. We changed this part of the text to better clarify the need for this study.
- Ln 73: in my mind, the structure of this sentence (as well as the manuscript) should be:
to evaluate the association of rs1049550 in new data and corroborate known results by adding new results in a meta-analysis.
to analyze different clinical sub-phenotypes in order to provide new evidence about the effects (estimates and directions) of rs1049550-T and, consequently, inform on therapy.
This sentence should be modified. Some words related to the importance of it should be reported in the discussion too.
Response: We modified this part of the text to better describe the aim of the paper following the reviewers suggested structure.
- Ln 138: OR, CI and P-value are related to the allelic model; it would be clearer if this is specified in the legend to avoid confusion with a genotypic model.
Response: We added the information to the legend of table 2.
Did the authors test genotypic models (dominant, recessive, etc)? This information could be interesting and would be reported, even if the result is not significant.
Response: We thank the reviewer for this suggestion and added test results for genetic models (additive, dominant, and recessive) underlying the association between the ANXA11 SNP and sarcoidosis and phenotypes of sarcoidosis in the new table 3 and added comments describing the findings in the results section paragraph 3.2 and the discussion.
- Ln 150 and 154 (Table3 and legend): authors reported rs1049550-T frequency is South Asians as 0.6619 while I found a frequency of 0.359 (from GnomAD v2.1.1:
https://gnomad.broadinstitute.org/variant/10-81926702-G-A?dataset=gnomad_r2_1),
please recheck this value.
Response: Indeed the South Asian population has a T allele frequency of 0.359 according to GnomAd v2.1.1, however the South Asian population encompasses especially populations from India and Pakistan. Most of the Chinese population is regarded as East Asian, therefore we mention the T allele frequency of the East Asian population in Table 4 and the legend of the table. We reworded the legend and text to better clarify the population and allele frequencies obtained from GnomAD and its 1KG database. In the 1KG database of East Asians, the Southern Han Chinese have a T allele frequency of 0.6619
The low allelic frequency observed in controls in Feng et al, more similar to South Asians than East Asians, should be due to any ascertainment bias, so excluding them from meta-analysis is a good point. Related to this, I think reporting somewhere heterogeneity statistics for the pooled OR =0.71 (Ln 149) would be important.
Response: This is indeed an important descriptor. The heterogeneity statistics are reported at the bottom of the plot in the new figure 1 and added to the text at line 183 of the revised version.
Also, I was wondering why Sikorova et al. 2020 study has not been included in this analysis, considering that the abstract reports:
“Further, ANXA11 rs1049550*A variant was associated with sarcoidosis (OR: 0.59 [0.39–0.89], p = 0.01).”
Response: Unfortunately, we could not include the Sikorova study in the meta-analysis because the genotype and allele frequencies were not provided in the manuscript. However we do refer to the study because of the association found, which is similar to previous findings.
- Ln 174 Table 4: for several studies, no frequencies but only sample sizes are reported (e.g. in Sikorova et al. 2020): were these data not available in the original study?
Response: Indeed in some of these studies the necessary data was not reported. We now added a footnote sign (†) to these studies in the table with clarification in the table legend.
In addition, it is not clear to me why in the Karakaya et al. results row, no comparison has been done between non-Löfgren’s syndrome patients and controls.
Response: We added the findings of our non-LS population to the table (Table 5).
- Table 4 and Figure 2: these items include too many results and I am not sure a unique table/figure would be the best way to represent such different phenotypes.
Apart from this remark, I think when multiple results are present for the same phenotypes, a meta-analysis should be applied to obtain an overall measure of rs1049550-T clinical impact. Being this aspect the major novelty of this paper (rather than the overall association with sarcoidosis), I think these data (especially new data) should be capitalized to the maximum (unless the authors have valid scientific reasons against this observation) and, mainly, observations at Ln 265-266 in the discussion should be proven.
Response: we agree with the reviewer: it was a lot of information that we tried to summarize. As one can see in the figure and the table, the clinical phenotypes and patient grouping for comparison, differed between the studies, which made it difficult to summarize. However, recapitulation should of course provide a clear overview. Therefore, we have now chosen to delineate the figure and table to the sarcoidosis phenotypes for which we provide new data. Thus, the staging results are transferred to the supplementary information, as is the summarizing figure 2 from the previous version.
Additionally, inspired by the comment of the reviewer, we decided to perform a meta-analysis for the sarcoidosis phenotypes for which we add new data: resolving and chronic disease, LS and non-LS patients. The results are presented in paragraph 3.4 and figure 2A and 2B.
And we agree with the reviewer that the message of our study is that rs1049550 associates with clinical phenotypes of sarcoidosis. We changed the manuscript accordingly.
- Ln 273: molecular data from Open Targets Genetics show that ANXA11 is the first candidate gene regulated by rs1049550, with for example rs1049550-T that increases ANXA11 expression on monocytes and in other tissues:
https://genetics.opentargets.org/variant/10_80166946_G_A
The discussion could include some references to this aspect.
Response : We thank the reviewer for pointing out this relatively new and interesting database. The results at Open Targets Genetics involve observations for rs1049550 that were imputed from respective lead variants from large aspecific GWAS studies that combine subject samples with medical databases. It is therefore interesting to see that indeed rs1049550 associates with sarcoidosis. The imputed association of rs1049550 with differential expression in monocytes is off considerable interest for the study of sarcoidosis. In the study of Javierre et al (that are incorporated at Open Targets, PMID: 27863249) the lead variant showed an effect on ANXA11 expression on monocytes and other cells.
We incorporated this and another study describing findings of ANXA11 in the lung in the revision at line 320-331.
- Ln 288-291: this content seems not pertinent in the conclusions, although I think, as I pointed out before, the authors should better discuss possible implications of their results in a clinical context.
Response: I want to apologize for not having payed attention to these last lines: these lines belonged to the template provided by ‘Cells’ and should have been erased.
Minor comments:
- Ln 52: Table3 is the first table cited, thus should be named as “Table1”; please renumber tables.
Response: we apologize for the mistake and inserted the correct numbers.
- Ln 146 and 149, p < 1·10-10: I think the p-value should be more informative if its exact value is reported (either because corresponds to a narrower 95% CI and thus a more precise OR estimation with respect to the best p-value obtained in single studies [Hoffman, OR=0.70 and p-value = 1.10-12], and because coherent with the modest Heterogeneity identified).
Response: We performed the meta-analysis with Revman (the meta-analysis programme of Cochrane), and Revman provides the approximate p value and a Z-score, from which we could seek the precise p-value that are now provided in the table.
- Ln 158 (Figure 1): as for the single study hits, the 95% CI should be plotted together with the combined OR.
Response: The meta-analysis program Revman provides a diamond at the bottom of the forest plot that represents the combined OR in the middle and covers its 95% CI at the tips of the diamond. We edited the figure and the legend to clarify this.
Reviewer 2 Report
The manuscript "ANXA11 rs1049550 Associates with Löfgren’s Syndrome and 2 Chronic Sarcoidosis Patients" by Karakaya et al. analyses the genotypes and allele frequencies in 262 Dutch sarcoidosis cases and compares them with 363 controls of the same origin. The manuscript is fairly written and the results are sound. I only have a few comments:
- The authors regret that due to "...contradictory findings the results have been difficult to oversee." However, in my view the "contradictory findings" result from two basic faults:
- Do not compare the incomparable! It is not astonishing that gene frequencies vary between populations. In my opinion it makes no sense to admix data from Han Chinese and African Americans with European data. Even in European populations you may find variations (see Lao 2008 doi: 10.1016/j.cub.2008.07.049)
- Sarcoidosis is a rare disease. Many studies suffer from the "error of small numbers", some studies more than others... From a statistical view, even a study including 262 sarcoidosis cases is a small study! Having this in mind I find it remarkable that the allele frequencies and OR in the studies involving European populations are relatively close. Again, I would disregard studies in non-European populations (see above).
- there are some minor typos in the text, e.g. unnecessary hyphens (e.g. gen-otyped (line 24); diffe-rence" (line 135))
Author Response
The manuscript "ANXA11 rs1049550 Associates with Löfgren’s Syndrome and 2 Chronic Sarcoidosis Patients" by Karakaya et al. analyses the genotypes and allele frequencies in 262 Dutch sarcoidosis cases and compares them with 363 controls of the same origin. The manuscript is fairly written and the results are sound. I only have a few comments:
The authors regret that due to "...contradictory findings the results have been difficult to oversee." However, in my view the "contradictory findings" result from two basic faults:
Do not compare the incomparable! It is not astonishing that gene frequencies vary between populations. In my opinion it makes no sense to admix data from Han Chinese and African Americans with European data. Even in European populations you may find variations (see Lao 2008 doi: 10.1016/j.cub.2008.07.049)
Sarcoidosis is a rare disease. Many studies suffer from the "error of small numbers", some studies more than others... From a statistical view, even a study including 262 sarcoidosis cases is a small study! Having this in mind I find it remarkable that the allele frequencies and OR in the studies involving European populations are relatively close. Again, I would disregard studies in non-European populations (see above).
Response: we thank the reviewer for the comment and agree totally. The “contradictory findings” referred to the results of the several studies that each analyzed a clinical phenotype but made different subdivisions, particularly with regard to Scadding stage studies. Although the results of these studies appear to contradict, they cannot be compared correctly. We added the text to clarify this: “ especially regarding the clinical phenotypes that were studied ” in line 251 of the revised version. Furthermore, we deleted findings for Scadding stage from the table and transferred these to the supplementary file.
Differences in allele frequency are often present between population. However, for many diseases it has been shown that despite these differences the direction of change in allele frequency in the disease population compared with healthy controls is similar between populations (PMID: 34907291, PMID: 26678098). For sarcoidosis this appears also true as we present in Figure.1 where all studies showed that the T.allele was less frequent in sarcoidosis than in controls.
Indeed, these are relatively small studies and the still allele frequencies and ORs in the studies are remarkably close to each other. For completeness and to show the worldwide association of rs1049550 with sarcoidosis we included the non-European populations. This shows that although population control frequencies for rs1049550 T-allele differ significantly between ethnic groups, in each population the T-alleles protects against development of sarcoidosis. As noted in line 184-186 (revised version) exclusion of the East-asian study, provided similar results. We added text to the discussion that combining data from different ethnic populations may be difficult.
there are some minor typos in the text, e.g. unnecessary hyphens (e.g. gen-otyped (line 24); diffe-rence" (line 135))
Response: we apologize for these typo’s and corrected them.
Round 2
Reviewer 1 Report
I would thank the authors for their great effort in improving their manuscript. So I support its publication. I would just ask to recheck the newly submitted version because Table 4 is not in a readable format.